# Reduction of Doppler and Range Ambiguity Using AES-192 Encryption-Based Pulse Coding [note 2]

**DOI:** 10.3390/s23052568

**Published:** 2023-02-25

**Authors:** Luke Kamrath, Michael Baginski, Scott Martin

**Affiliations:** 1Department of Electrical and Computer Engineering, Auburn University, 200 Broun Hall, Auburn, AL 36849, USA; 2Department of Mechanical Engineering, Auburn University, Auburn, AL 36849, USA

**Keywords:** radar, random, pseudo-random, BPSK, encryption, range-ambiguity, Ipatov, Barker, CLEAN

## Abstract

This research investigates the use of a Binary Phase Shift Key (BPSK) sequence derived from the 192-bit key Advanced Encryption Standard (AES-192) algorithm for radar signal modulation to mitigate Doppler and range ambiguities. The AES-192 BPSK sequence has a non-periodic nature resulting in a single large and narrow main lobe in the matched filter response but also produces undesired periodic side lobes that can be mitigated through the use of a CLEAN algorithm. The performance of the AES-192 BPSK sequence is compared to an Ipatov–Barker Hybrid BPSK code, which effectively extends the maximum unambiguous range but has some limitations in terms of signal processing requirements. The AES-192 based BPSK sequence has the advantage of having no maximum unambiguous range limit, and when the pulse location within the Pulse Repetition Interval (PRI) is randomized, the upper limit on the maximum unambiguous Doppler frequency shift is greatly extended.

## 1. Introduction

The inherent Range and Doppler ambiguities in radar signal processing are significant issues that cause uncertainties in the location and relative velocity of targets. There are various methods available for mitigating these ambiguities, including using multiple subantennas [1], MIMO with pulse coding [2], beam pattern masking [3], dual phase term tracking [4], the use of multiple hypothesis tracking models and data association [5], mitigating range ambiguity using Doppler Division Multiple Access (DDMA) [6] and the use of phase codes to modulate the baseband signal [7].

The work presented here is an extension of our earlier works [8,9] and focuses on extending the maximum unambiguous range and eliminating the Doppler ambiguity by using an encryption based Binary Phase Shift Keying (BPSK) sequence. This is related to the work done by Levonon [7] which investigated BPSK based on Ipatov codes that extend the maximum unambiguous range. The reason Levonon chose Ipatov codes is that they are known to have good correlation properties when used with a specialized mismatched filter, which can greatly improve the performance of the radar system without the need of additional hardware. However, there are limitations using this approach, including the complexity of implementing the sequences, CPI duration constraints, and diminished target identification when Doppler shifts are present. This work additionally demonstrates that the inherent randomness of the encrypted signal allows for a randomized Pulse Repetition Interval (PRI), effectively extending the maximum observable unambiguous Doppler frequency with minimal reduction in performance.

The research presented here uses AES-192 Encryption, the 192-bit key variant of the Advanced Encryption Standard [10], to produce BPSK sequences that can mitigate the inherent range ambiguity observed in a radar signal. Since the encrypted signal is uncorrelated with past or future signal transmissions, only the true target location will cause a peak in the matched filter’s response. Apart from BPSK, there are other Phase Shift Keying (PSK) modulation techniques that can also be used for uniquely coding radar pulses. Quadrature Phase Shift Keying (QPSK) is more robust than BPSK, as it allows for four different signal states rather than two; 8-PSK and 16-PSK are also possible, but are generally less common due to their higher complexity. Using an encryption based sequence for PSK modulation requires hardware capable of generating the encrypted bits in real time.

The PSK method used determines how many encrypted bits map to each code chip. The BPSK code used in this work can be directly mapped from one encrypted bit to one BPSK chip; however, codes with more than two phase states such as QPSK or 8PSK require multiple encrypted bits per chip, thus increasing the rate at which the hardware must generate encrypted bits. A functioning Software Defined Radio (SDR) implementation of this radar on a device such as a Field Programmable Gate Array (FPGA) would require an increasing amount of FPGA resources as chipping frequencies and PSK states are increased. A lowest cost implementation would be best suited to a BPSK code. The use of BPSK coding in this work was chosen for two reasons. Firstly, the prior work in range ambiguity mitigation [7] to which the authors are comparing their results was created using BPSK coding. Secondly, the authors have more experience with encryption and BPSK implemented in SDR hardware and plan to test a functional version of the radar, which makes it a more suitable choice than other coding schemes.

This research further demonstrates that a second AES-192-based sequence can be used to randomly locate the pulse of width τ within a PRI. RPL effectively mitigates the Doppler ambiguity observed in a radar signal. Earlier research that specifically focused on reducing the Doppler ambiguity in automotive radar was done by Gonzalez et al. [11]. Their method focused on resolving the Doppler ambiguity by using Multiple-Input Multiple-Output (MIMO) arrays and Time Division Multiplexing (TDM). Multi-element antenna arrays add additional physical system complexity and costs. In this work, a monostatic radar is simulated with separate transmit and receive antennas; therefore, only two antenna elements are required. There is still additional complexity in the signal processing, but the overall RF hardware costs are reduced by using fewer antennas.

## 2. AES-192 Pseudo-Random Generator

This work is an extension of our earlier work that used AES-192 encryption [10] to modulate a baseband radar signal. Here, we use AES-192 encryption to modulate a baseband radar signal and introduce the idea of randomly shifting the “on” time of the pulse within a PRI as a way of reducing Doppler ambiguity. The AES-192-based sequence would in theory eliminate range ambiguity, since the entire baseband pulse sequence is uncorrelated. Prior radar work that used encrypted phase codes was done by Shahrab and Soleimani as a means of preventing radar jamming and spoofing [12]. The work presented here uses encrypted phase codes to modulate a very large number of PRIs and effectively extend the ambiguous range to the limit of processable and storable data by the radar. The AES-192-based sequence used to modulate the baseband radar signal can be repeatably started at any predefined time, and if desired, additional encrypted information can easily be embedded in the signal without penalty if a need arises. AES-192 encryption was selected based on its computational efficiency and the author’s familiarity with the algorithm. Other methods of encryption will likely produce similar results, but this research only considers the AES-192 encryption algorithm.

All non-encrypted data are referred to as plain-text (PT), and encrypted data are referred to as crypto-text (CT). The CT is produced by encrypting the PT with a binary key and is then used to modulate the baseband radar signal. The number 192 in the AES-192 encryption designation simply indicates that the encryption key is 192-bits in length. A single encryption cycle of AES-192 will produce 128-bits (one block) of the binary phase code. In a continuous transmission mode, the encryption needs to be done at the rate of the BPSK chipping frequency (modulation rate) divided by 128. The complete AES-192 encryption algorithm is described in [10], and its inputs and output are abstracted by Equation (Equation 1).
(1)$$CT_{128}=AES_{192}\left(PT_{128},key_{192}\right)$$

The PT blocks are changed after each encryption cycle to produce pseudo-random behavior by including the current block count in the PT field. The PT blocks to be encrypted are found using Equation (Equation 2), where dec2bin converts PT from a decimal base-10 to a binary base-2 representation, *k* is the encryption block count, η is a user specified prime number that promotes bit diversity in the plain text, and *l* is a user-specified fill value. The maximum length of a unique sequence produced in this manner by a single key is $2^{135}\approx 4e40$ BPSK chips.
(2)$$PT_{128}=dec2bin\left(k\eta +l,128\right)$$
The PT includes the block count of each encryption cycle and can be used to obtain a time of transmit by back solving for the block count and pulse number after correlation.

## 3. Pulse Code

The periodic times during which the radar is actively transmitting can be easily expressed in terms of time *t*, the active period of the pulse τ, and the duration of the pulse *T*. Equation (Equation 3) defines the gate function that represents the active period of a pulsed radar signal with a constant PRI.
(3)$$gate\left(t\right)=\left\{\begin{array}{cc} 1.0, & (t\;\mathrm{mod}\;T)<\tau \\ 0, & otherwise \end{array}\right.$$

Modulating the signal with CT bits requires a means of selecting bits as a function of time *t*. The CT bit *n* and encryption block *k* contain several prerequisite variables that must first be defined. The current pulse *P* (integer) calculated by Equation (Equation 4) is used later to offset the chip index by the number of chips per pulse.
(4)$$P\left(t\right)=floor\left(t/T\right)$$

The time within a pulse *p* is calculated according to Equation (Equation 5). This is used later to calculate the waveform characteristics, and it ranges in value from zero to *T* due to the modulus function, resembling a saw tooth wave.
(5)p(t)=tmodT

The unmodulated baseband signal has a period *T* and is “on” for a duration τ. The fixed number of chips used to modulate the pulse while it is “on” are determined according to Equation (Equation 6) based on the chipping frequency $f_{c}$, or the equation can be inverted to obtain $f_{c}$ for a desired number of chips per pulse.
(6)$$P_{N}=\tau f_{c}$$

In this research, $P_{N}$ is set to 13 exactly; this constrains the value of either τ or $f_{c}$. Equations (Equation 4)–(Equation 6) are used to produce Equations (Equation 7) and (Equation 8), the current block and bit index, respectively, as a function of *t*.
(7)$$k\left(t\right)=floor\left(\frac{P\left(t\right)P_{N}}{128}+\frac{p\left(t\right)P_{N}}{128\tau }\right)$$
(8)$$n\left(t\right)=floor\left(P\left(t\right)P_{N}+\frac{p\left(t\right)P_{N}}{\tau }\right)\;\mathrm{mod}\;128$$

It should be noted that Equation (Equation 7) and Equation (Equation 8) will only produce the correct indices during the active (non-zero) portion that occurs for τ seconds, and the values during the inactive period of the pulse are multiplied by zero due to the gate function described by Equation (Equation 3). Using the block and bit at any time *t*, the current CT bit can be obtained according to Equation (Equation 9). Here, the bracketed [n(t)] represents the array bit index within the kth 128-bit CT block.
(9)$$bit\left(t\right)=AES_{192}\left(dec2bin\left(k\left(t\right)\eta +l,128\right),key\right)\left[n\left(t\right)\right]$$

Equation (Equation 9) is then used to generate the phase-coded high-frequency $f_{0}$ carrier waveform of the AES-192-based constant PRI signal according to Equation (Equation 10).
(10)$$V_{RF}\left(t\right)=\left(2\times bit\left(t\right)-1\right)cos\left(2\pi f_{0}t\right)gate\left(t\right)$$

## 4. Random Pulse Location within a PRI

One additional feature was added to the AES-192 modulation scheme to address the problem of Doppler ambiguity. By randomly locating the “on” portion of the pulse within each PRI, the effective PRI is changed in a random manner, which significantly reduces the Doppler ambiguity. To achieve this, a second encryption-based sequence is used to determine the activation time delay before the “on” portion of the pulse. This has the effect of randomizing the PRI/PRF according to Equation (Equation 11). Here, $\tau_{0}\left(t\right)$ defines the encryption-based random pulse activation time delay before the onset of the pulse for every PRI.
(11)$$\tau_{0}\left(t\right)=\frac{T-\tau }{2^{P_{B}}}\sum_{n=0}^{P_{B}-1}crypto\left[n+P\left(t\right)P_{B}\right]\times 2^{n}$$
Generating the RPL signal is done using a modified version of Equation (Equation 10) that delays each pulse activation point according to $\tau_{0}\left(t\right)$ in Equation (Equation 12). A sample of RPL for several pulses is shown in Figure 1 for reference.
(12)$$V_{RPL}\left(t\right)=\left\{\begin{array}{cc} \left(2\cdot bit\left(t-\tau_{0}\left(t\right)\right)-1\right)cos\left(2\pi f_{0}t\right), & \left(p\left(t\right)\geq \tau_{0}\left(t\right)\right)\&\left(p\left(t\right)\leq \tau +\tau_{0}\left(t\right)\right)\\ 0, & otherwise \end{array}\right.$$

Table 1 shows the pattern of generator function bit index integer ranges for several PRI (*T*) time intervals, where P(t) is Equation (Equation 6), the current pulse count as a function of *t*; n(t) is Equation (Equation 8), the current encryption block’s bit index as a function of *t*; and k(t) is Equation (Equation 7), the current encryption block in use as a function of *t*. Note that dashes are used to show ranges of values and commas are used to separate multiple values when the equation output changes during the PRI time interval. During the time range from pulse 9 to 10, the current encryption block’s bit index n(t) exceeds the 128 bit modulus and rolls back to zero at the same time the current encryption block k(t) index increments from zero to one. Essentially, every pulse, 13 CT bits are used, and every 128 CT bits, a block is used.

## 5. CLEAN Algorithm

When several targets are present in the radar return signal (rrs(t)), the target causing the largest peak in the matched filter’s response tends to eclipse or obscure the smaller or weaker targets. When AES-192 encryption is used, there is the additional problem of low-level noisy side lobes appearing in the matched filter’s response that need to be removed. Therefore, a “CLEAN” algorithm is used to help remove the unwanted noisy side lobes and resolve the less significant targets in the matched filter response (mfr(t)). The CLEAN algorithm was first used in radio astronomy [13,14] to deconvolve star images that were assumed to be point sources. The algorithm is adapted here for radar signal processing. The algorithm first identifies the location and approximate size of every observable target in the matched filter response (mfr(t)) above a predefined threshold. Next, the position and location of the target causing the largest peak in the matched filter’s response are used to generate the return signal that would occur for only that target. This signal is referred to as $x_{0}^{\prime }\left(t\right)$, and it is subtracted from the initial received radar return signal designated $rrs_{0}\left(t\right)$. This effectively removes or erases the contribution of a the target causing the largest peak, resulting in a new data record referred to as $rrd_{1}\left(t\right)=rrd_{0}\left(t\right)-x_{0}^{\prime }\left(t\right)$. This improves the detectability of weaker or lower contrast targets that may be hidden or eclipsed by stronger or more dominant returns. Some versions of CLEAN can also help to reduce clutter and improve the overall performance of the radar system. The matched filter’s response is then reconstructed using $rrd_{1}\left(t\right)$, and previously eclipsed targets may then be observed. The largest target remaining in the matched filter’s response after the first iteration of CLEAN is then identified and its location and approximate size catalogued. Its radar return signal is then created $x_{1}^{\prime }\left(t\right)$ and subtracted from the previous data record producing $rrd_{2}\left(t\right)=rrd_{1}\left(t\right)-x_{1}^{\prime }\left(t\right)$, and $rrd_{2}\left(t\right)$ is used create the new matched filter response and the process repeated until all identifiable target locations and sizes are recorded and catalogued. The use of AES-192 encryption adds an additional requirement that for every target removed by the CLEAN algorithm, the matched filter’s response must be regenerated. Unlike sequence CLEAN [15], the simplified CLEAN algorithm used here assumes the largest peak in the ambiguity function is due to an actual target, and thus no branching tree is used to resolve constructive side lobe interference. Future work in this area is planned to determine the most optimal CLEAN algorithm.

A block diagram of the simplified CLEAN algorithm used is shown in Figure 2 and is outlined in the steps listed below.
The process begins by generating the matched filter’s response to the $rrs_{n}\left(t\right)$ and identifying the target with the largest peak.The identified target’s complex amplitude, range and Doppler frequency (target parameters) are obtained from the matched filter bins and saved to a list of identified targets (target memory).Using the target parameters, a simulated target’s return signal is generated $x_{n}^{\prime }\left(t\right)$ (negation replica).This simulated return signal $x_{n}^{\prime }\left(t\right)$ is then subtracted from the initial $rrs_{n}\left(t\right)$, resulting in a new CLEANED signal $rrs_{n+1}\left(t\right)$ that is stored in the processing buffer, thereby overwriting the previous signature with one fewer target present.If any significant target is identified in the new matched filter response, the above process is repeated. Otherwise, when no new targets are identified, the procedure stops and the algorithm goes to step 6.At this point, the target memory can be exported or an image can be created using the identified target information. The target memory can also be used to create an ideal matched filter response for each target, which may be superimposed to produce a “cleaned” image.

## 6. Ipatov-13 Barker-13 Hybrid

Prior work in the area of mitigating or reducing range ambiguity was done by Levanon [7] using Ipatov inter-pulse and Barker intra-pulse BPSK codes. Levanon’s method of mitigating range ambiguity was successful, and his results can be directly compared to the results of this work. In order to perform the comparison fairly, a Barker-13 sequence is used for the intra-pulse BPSK code. The Barker-13 sequence allows the spectral bandwidth and average power of both techniques to be equal, allowing a fair comparison of each method’s effectiveness at reducing the range ambiguity. Ipatov codes extend the effective unambiguous range by the Ipatov code length (e.g, if the unambiguous range is $R_{u}$, with an Ipatov code length of 13, the new unambiguous range = 13$R_{u}$). An additional characteristic of Ipatov codes is their “ideal” periodic correlation response that results when a specialized miss-matched filter is used (this technique is discussed in [7]). The sequence length of the miss-matched filter is found by taking the length of the CPI and subtracting the 2× Ipatov pulse sequence length from it.

The phase states for each sequence used are provided in Table 2. The length of the CPI needs to be an integer multiple of the Ipatov–Barker hybrid (H-BPSK) length, and the encrypted signal (AES-192) length will be identical. Furthermore, the same CLEAN algorithm is used on both signals prior to the comparison.

## 7. Simulation Results

### 7.1. Constant Pulse Location in PRI

The Ipatov-13 and Barker-13 Hybrid codes were used to simulate the radar return signal from multiple targets using several different CPIs. The results shown here are for the 78 pulse CPI simulations. Figure 3 is a plot of the power spectral density versus a normalized frequency $f_{n}$ ($f_{n}=\frac{f}{f_{c}}$) for the Ipatov–Barker 13 and AES-192 coding, for both constant and RPL, confirming they have nearly identical power spectral density signatures. Both signals have nearly identical pulse parameters, with the exception being that the specific phases of the BPSK chips are different. A 10% duty cycle (τ=0.1T) was used and the chipping frequency was adjusted to have exactly 13 BPSK chips over the interval τ ($f_{c}=13/\tau$). An intermediate frequency ($f_{IF}=10f_{c}$) was used for the quadrature simulation and Doppler processing. The sampling frequency was double the minimal Nyquist frequency required ($f_{s}=40f_{c}$).

A comparison of the matched filter’s response found using Levanons’s Ipatov–Barker codes and the AES-192-based sequence developed in this work is shown in Figure 4. A broken horizontal axis is used to highlight the important details observed in the responses, and the targets are identified by red arrows. The matched filter’s response for two targets having time delays of 0.0*T* and 0.3*T* is simulated and shown before the CLEAN algorithm is used in (a)—upper portion of Figure 4a—and after CLEAN in (b)—lower portion of Figure 4. The target located at zero is treated as a virtual target and used to normalize the output of the matched filter’s response. It should be noted that at the 13th pulse interval the entire matched filter output for the Ipatov–Barker code begins to repeat itself, and this continues every 13th pulse. This is expected since the Ipatov-13 sequence will cause the effective unambiguous range to be extended by a factor of 13. The matched filter’s response found using the AES-192 derived sequence does not repeat at regular intervals. However, low-level noisy side lobes do appear in the matched filter’s response and are approximately 22 dB below the peak.

Figure 4b shows the matched filter output for both signals after the CLEAN algorithm is applied. It is obvious that the noisy side lobes are significantly reduced by approximately 25 dB, and the targets are clearly visible. Furthermore, the periodic elements of the AES-192 signal’s matched filter are effectively mitigated by the CLEAN algorithm.

The second comparison, shown in Figure 5, is of the same Ipatov–Barker and AES-192 signals as before but for five targets. The targets located at time delays of 0.0*T* and 0.3*T* were kept from the previous simulation, and an additional three very tightly grouped targets with time delays of 1.1*T*, 1.115*T*, and 1.13*T* were added. These targets are separated by less than the theoretical range resolution defined by the pulse width τ and wilcause their target responses to overlap in the matched filter’s output. This was done to investigate how the CLEAN algorithm would perform when processing overlapping targets. The matched filter’s output clearly shows the previous two targets. However, the additional three closely grouped targets appear as a single target in the matched filter response prior to the application of the CLEAN algorithm. After CLEAN is applied, the additional targets visually appear as a single target in the matched filter response plot. However, the CLEAN algorithm did identify the three tightly grouped targets as separate targets even though they are separated by less than τ in the matched filter’s response. It should be noted that the matched filter signature for the Ipatov-13 sequence repeats itself every 13 pulses as before, and this is expected.

In Figure 5a, the additional three targets are obscured in the matched filter output of the AES-192 signal by the repetitive range lobes as before. In Figure 5b, CLEAN is applied to the original signals, and the output of the matched filter reveals the previously obscured targets. Even though they do appear as a single target at ∼1.1 (normalized time) in this figure, later results show they are fully separable.

### 7.2. Random Pulse Location within PRI

The next phase of the research investigated using encryption-based Random Pulse Location (RPL) for pulses of width τ within a PRI, as shown in Figure 1. Introducing a level of randomness to the pulse location within a PRI has the effect of randomizing the PRI and PRF of the radar and minimizes the Doppler ambiguity that typically occurs when the Doppler frequency shift $f_{d}$ is greater than $\pm \frac{PRF}{2}$. It is important to note that this technique will only work if the width of the pulse, τ is sufficiently small compared to the PRI. If the pulse width is too large, it will not effectively randomize the PRF and may not help to resolve the Doppler ambiguity. Levanon’s work only addressed the range ambiguity mitigation properties of the Ipatov and Barker BPSK codes. This research includes the addition of RPL on Ipatov–Barker signals to show the effects of randomizing an ideal code such as an Ipatov code. It is expected that the random component of RPL will cause detrimental effects on the ideal correlation properties of the Ipatov code. The AES-192 based code, however, is already random, and through the use of a CLEAN algorithm, the side lobes of the signal can be greatly reduced.

The matched filter’s response for the RPL AES-192-based sequence using the previous two target scenarios is shown in Figure 6. The RPL Ipatov–Barker matched filter’s response is also shown. The top portion of Figure 6 compares responses prior to the CLEAN algorithm being applied on both signals. Here, the second smaller target’s response is eclipsed in the RPL AES-192 response. However, after the CLEAN algorithm is applied to the RPL AES-192 response, the second target’s signature becomes clearly visible and the plateau-like noisy side lobes are reduced by ∼50 dB.

The matched filter’s response using the RPL AES-192-based sequence for the earlier five target scenario is compared to the RPL Ipatov–Barker response in Figure 7. After the CLEAN algorithm is applied, the two larger targets are clearly visible, and the three closely grouped targets are observable, even though it is still difficult to distinguish each individual target. Figure 8 provides a close up on the three tightly grouped targets placed at ∼1.1 (normalized time) in the five target simulation. With the application of the CLEAN algorithm, the buried targets become fully observable.

### 7.3. Doppler Analysis

In addition to testing the range resolving performance of the Ipatov–Barker code and the AES-192-based sequence with and without the CLEAN algorithm, each pulse modulation scheme’s ability to resolve Doppler information was also investigated. To do this, a virtual target at zero range and zero Doppler shift with an amplitude of 1.0 was used. Each modulation method was used to create a zero-Doppler slice and zero-delay slice of the sampled ambiguity function. The target delay is plotted in time normalized by PRI ($t_{n}=\frac{t}{T}$) for visual clarity. Figure 9 uses the Ipatov–Barker sequence to generate the zero-Doppler slice and zero-delay slice (thick dark blue line) of the ambiguity function. In addition to this information, the maximum value of the ambiguity function at a given range is shown in the top portion and maximum value for a given Doppler shift in the lower portion of the Figure 9. The important features to note are that the Doppler signature is periodic with a repetitive period of ($1.0\;f_{d}T$), indicating that the maximum unambiguous Doppler shift is still limited to $fd_{max}=\pm \;\frac{PRF}{2}$. The large phantom target that appears at $t_{n}=1$ is an artifact. This phantom target is not at zero velocity and does not appear in the zero-Doppler slice. The Ipatov Barker encoding method did manifest all the salient features observed in the ambiguity function plot provided in [7].

Figure 10 uses the AES-192-based BPSK sequence to generate the zero-Doppler slice and zero-delay slice (thick dark blue line) of the ambiguity function. The maximum value of the ambiguity function at a given range is shown in the top portion and the maximum value at a given Doppler shift in the lower portion of Figure 10. The Doppler signature shows the same periodicity as observed using Ipatov–Barker encoding with a repetitive period of ($1.0\;f_{d}T$), indicating that the maximum unambiguous Doppler shift is still $fd_{max}=\pm \;\frac{PRF}{2}$. The phantom target does appear at $t_{n}=1$, but its peak is not as sharp and does not contain a replica of the main lobe. Overall, the encrypted BPSK encoding did identify all the typical features observed in an ambiguity function plot. Additionally, the main lobe (full target replica) range ambiguity is also mitigated at $t_{n}=13$.

Figure 11 uses RPL with the Ipatov–Barker sequence to generate the zero-Doppler slice and zero-delay slice (thick dark blue line). The maximum value of the ambiguity function at a given range is shown in the top portion and the maximum value at given Doppler shift in the lower portion of the Figure 11. The typical Doppler periodicity seen at $fd_{max}=\pm \;\frac{PRF}{2}$ is not present, indicating that the Doppler ambiguity over the frequency range shown has been mitigated by introducing RPL into the Ipatov–Barker sequences. The phantom target at $t_{n}=1$ has also been eliminated, but a plateau-like noise floor has been introduced by RPL. This shows the disrupting effect of RPL on the ideal correlation properties of the Ipatov code.

Figure 12 uses the same RPL delays with the AES-192-based sequence to generate the zero-Doppler slice and zero-delay slice (thick dark blue line). The maximum value of the ambiguity function at a given range is shown in the top portion and the maximum value at given Doppler shift in the lower portion of Figure 12. The typical Doppler periodicity seen at $fd_{max}=\pm \;\frac{PRF}{2}$ is not present, indicating that the Doppler ambiguity over the frequency range shown has been mitigated by introducing RPL into the AES-192 coded sequences. The phantom target at $t_{n}=1$ has also been eliminated.

## 8. Conclusions

In this research, the abilities of the Ipatov–Barker, AES-192, RPL Ipatov–Barker and RPL AES-192 sequences to resolve target location and mitigate range and Doppler ambiguity with and without the use of the CLEAN algorithm were investigated. The CLEAN algorithm is a technique used to reduce the noise in the matched filter’s response. It works by iteratively identifying and subtracting a model of each target from the data while preserving the target signals. As the study found, after applying the CLEAN algorithm, the side lobes are reduced by approximately 25 dB, which means that the CLEAN algorithm is effective at removing much of the noise and greatly improves the system’s ability to detect and locate targets accurately.

It was found that before the CLEAN algorithm was applied to any of the signals with random components—AES-192, RPL Ipatov–Barker and RPL AES-192—these methods developed undesirable side lobe noise in the matched filter’s response. This noise can potentially eclipse weaker targets and may even appear as false targets if not removed. However, after the CLEAN algorithm is applied, the noisy side lobes are reduced by approximately 25 dB, as shown in Figure 4, Figure 5, Figure 6, Figure 7 and Figure 8.

Using the AES-192 encoding, each transmitted pulse has a set number of phase chips that are uncorrelated with previous or future pulses. Given a sufficient number of phase chips within each pulse and a coherent processing interval that spans multiple pulses, the standard range ambiguity that appears in typical radar signal processing was mitigated. The negative attributes of the AES-192-based sequence that manifest in the matched filter’s response as spurious side lobe noise require an additional processing algorithm to make the signal viable. Using the encrypted BPSK sequences in combination with the CLEAN algorithm, the target appears only at the true round trip delay time in the matched filter response with minimal side lobe noise. The primary drawback of this method is the processing required to perform the CLEAN algorithm. The processing may add latency to target detection and increase the total cost of the radar system.

In addition to examining the range resolving performance of the four modulation sequences (Ipatov–Barker, AES-192, RPL Ipatov–Barker, RPL AES-192), the ability of each to resolve Doppler frequency information was also investigated with and without the CLEAN algorithm. The effectiveness of the modulation schemes was compared by plotting the zero-Doppler slice and zero-delay slice of the sampled ambiguity functions of each method. The results showed that by randomly locating a pulse within the PRI, using either AES-192 or Ipatov–Barker encoding, the Doppler spectrum replication that occurs for Doppler frequencies beyond $f_{d}=\pm \frac{1}{2PRI}$ was eliminated. Hence, there was no maximum unambiguous Doppler frequency limit observed for the frequency range considered.

## Figures and Tables

**Figure 1 sensors-23-02568-f001:**
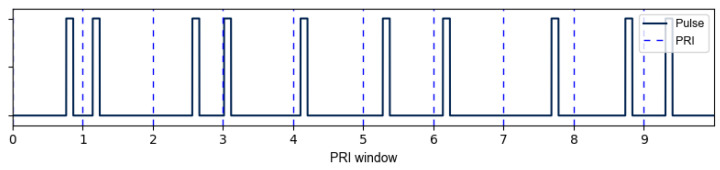
Sample RPL gate function activation delays $\tau_{0}\left(t\right)$ to achieve a random PRI.

**Figure 2 sensors-23-02568-f002:**
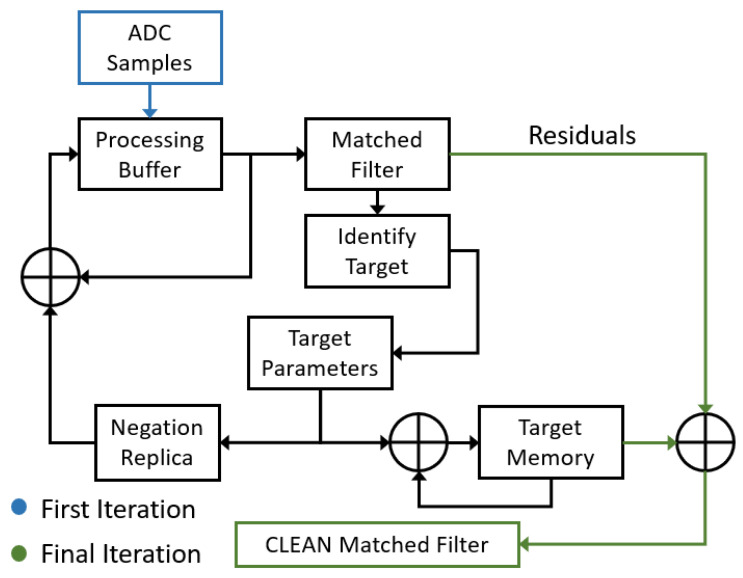
Simplified block diagram of a CLEAN algorithm.

**Figure 3 sensors-23-02568-f003:**
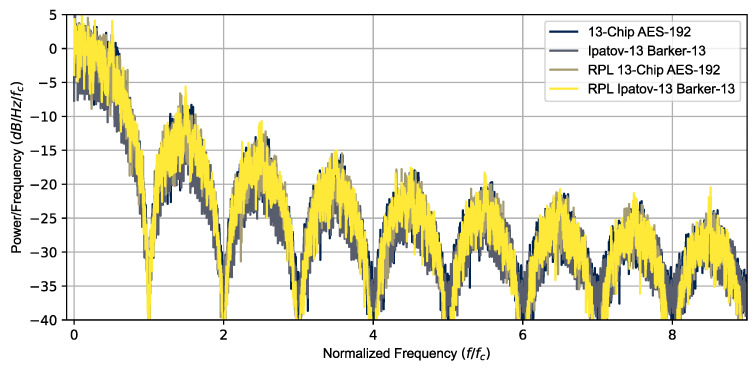
Power spectral density overlay for all simulated signals. All the signals have similar PSDs.

**Figure 4 sensors-23-02568-f004:**
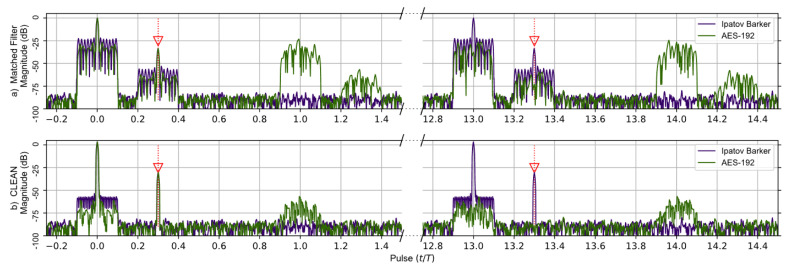
Ipatov–Barker method (Purple) and AES-192 method (Green) matched filter’s response for 78 simulated pulses. (**a**) Top, matched filter’s response before CLEAN. (**b**) Bottom, CLEAN Reconstructed matched filter’s response when applied to both targets. The Ipatov–Barker method begins to repeat after pulse 13, but the AES-192 method does not. The side lobes of the AES-192 method are reduced by CLEAN.

**Figure 5 sensors-23-02568-f005:**
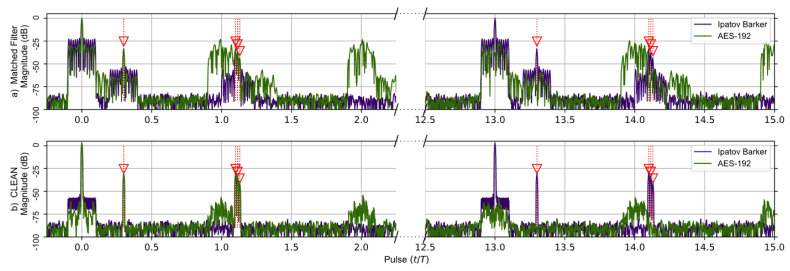
Ipatov–Barker method (Purple) and AES-192 method (Green) matched filter’s response for 78 simulated pulses. (**a**) Top, matched filter’s response before CLEAN. (**b**) Bottom, CLEAN Reconstructed matched filter’s response when applied to all targets. The Ipatov–Barker method begins to repeat after pulse 13, but the AES-192 method does not. The side lobes of the AES-192 method are reduced by CLEAN.

**Figure 6 sensors-23-02568-f006:**
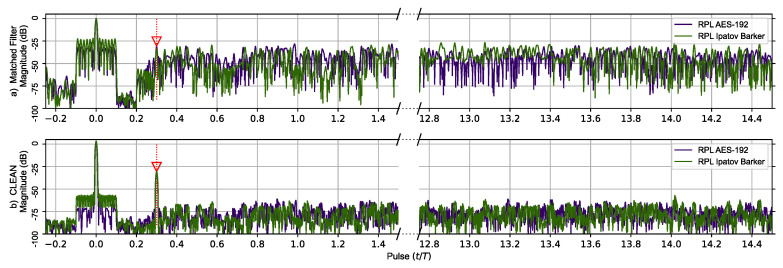
RPL AES-192 method (Purple) and RPL Ipatov–Barker method (Green) matched filter’s response for 78 simulated pulses. (**a**) Top, matched filter’s response before CLEAN. (**b**) Bottom, CLEAN reconstructed matched filter’s response when applied to both targets. The side lobes of the virtual target at zero eclipse the second target until CLEAN is used to reduce them.

**Figure 7 sensors-23-02568-f007:**
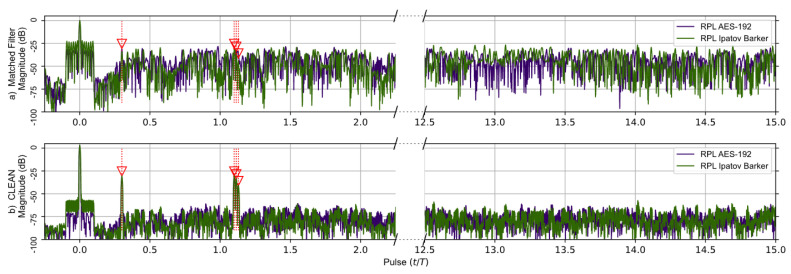
RPL AES-192 method (Purple) and RPL Ipatov–Barker method (Green) matched filter’s response for 78 simulated pulses. (**a**) Top, matched filter’s response before CLEAN. (**b**) Bottom, CLEAN reconstructed matched filter’s response when applied to all targets. The noisy side lobes are significantly reduced by CLEAN revealing all buried targets.

**Figure 8 sensors-23-02568-f008:**
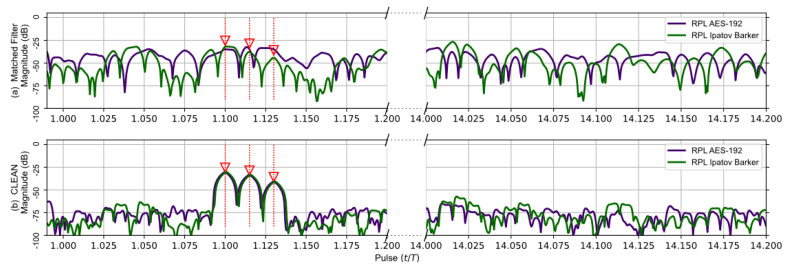
Duplicate data from Figure 7; focused on the tight group of three targets buried in the side lobes before CLEAN. The CLEAN algorithm was able to recover the buried targets.

**Figure 9 sensors-23-02568-f009:**
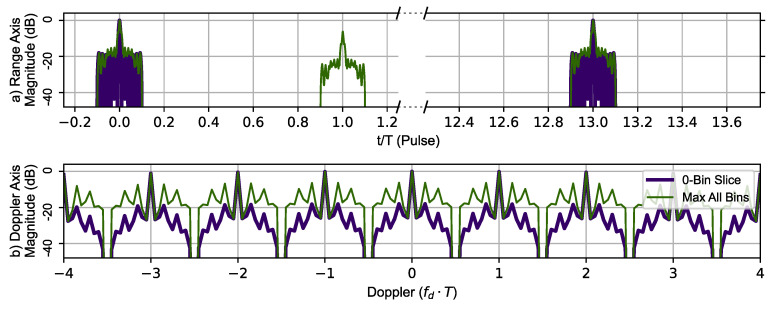
Ipatov–Barker method’s 78 pulse ambiguity function cross-sections in peak relative dB scale. (**a**) Top, zero Doppler cross-section (Purple), Max all Doppler cross-sections (Green). (**b**) Bottom, zero delay cross-section (Purple), Max all delay cross-sections (Green). The Doppler slice of the ambiguity function contains repeated peaks corresponding to the Doppler ambiguity.

**Figure 10 sensors-23-02568-f010:**
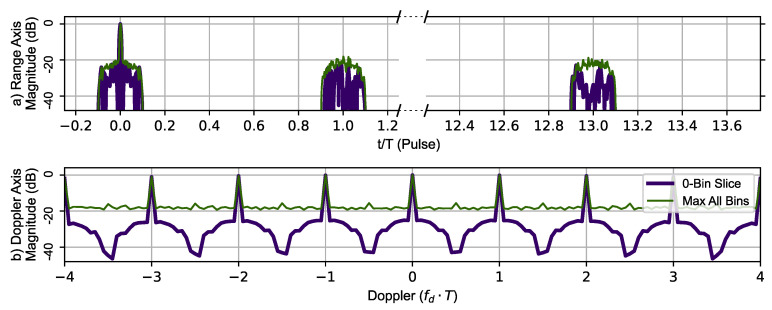
AES-192 method’s 78 pulse ambiguity function cross-sections in peak relative dB scale. (**a**) Top, zero Doppler cross-section (Purple), Max all Doppler cross-sections (Green). (**b**) Bottom, Zero delay cross-section (Purple), Max all delay cross-sections (Green). The Doppler slice of the ambiguity function contains repeated peaks corresponding to the Doppler ambiguity.

**Figure 11 sensors-23-02568-f011:**
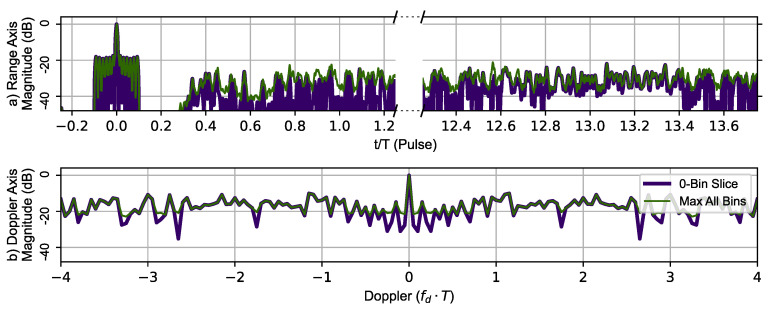
RPL Ipatov–Barker method’s 78 pulse ambiguity function cross-sections in peak relative dB scale. (**a**) Top, zero Doppler cross-section (Purple), Max all Doppler cross-sections (Green). (**b**) Bottom, zero delay cross-section (Purple), Max all delay cross-sections (Green). There are no duplicate peaks in the Doppler slice of the ambiguity function.

**Figure 12 sensors-23-02568-f012:**
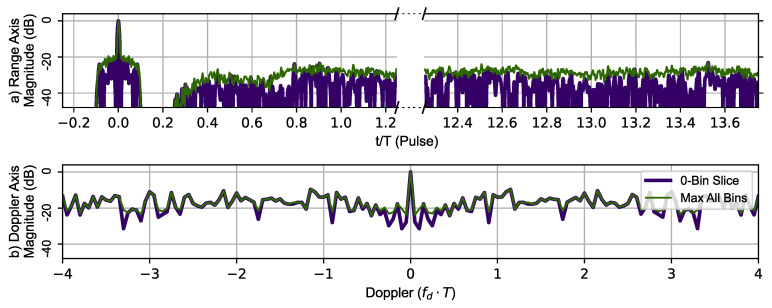
RPL AES-192 method’s 78 pulse ambiguity function cross-sections in peak relative dB scale. (**a**) Top, zero Doppler cross-section (Purple), Max all Doppler cross-sections (Green). (**b**) Bottom, zero delay cross-section (Purple), Max all delay cross-sections (Green). There are no duplicate peaks in the Doppler slice of the ambiguity function.

**Table 1 sensors-23-02568-t001:** Generator function bit index integer ranges for several PRI (*T*) time intervals.

Time *t*	P(t)	n(t)	k(t)
0−T	0	0–12	0
T−2T	1	13–25	0
⋮	⋮	⋮	⋮
9T−10T	9	117–127, 0–1	0, 1
10T−11T	10	2–14	1

**Table 2 sensors-23-02568-t002:** Various length 13 BPSK codes.

Code	Phase States												
Barker-13	+1	+1	+1	+1	+1	−1	−1	+1	+1	−1	+1	−1	+1
Ipatov${}_{T}$-13	+1	−1	−1	+1	−1	+1	+1	+1	+1	+1	−1	+1	+1
Ipatov${}_{C}$-13	+2	−3	−3	+2	−3	+2	+2	+2	+2	+2	−3	+2	+2

## Data Availability

Not applicable.

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
