# Peer review of "Reduction of Doppler and Range Ambiguity Using AES-192 Encryption-Based Pulse Coding†"

_sensors, 2023, doi:10.3390/s23052568_

Round 1

Reviewer 1 Report

This manuscript sensors-2147583 investigates the use of encrypted pseudo-random binary phase shift (AES-192) sequences for radar signal modulation to mitigate Doppler and range ambiguities. These AES-192 BPSK sequences are generated using the AES-192 encryption algorithm and have a non-periodic nature, resulting in a single large and narrow main lobe in the matched filter response, but also producing periodic side lobes that need to be removed using the CLEAN algorithm. The performance of the AES-192 sequences is compared to Ipatov-Barker Hybrid BPSK (H-BPSK) sequences, which effectively extend the maximum unambiguous range, but have some limitations in terms of signal processing requirements. The AES-192 sequences have the advantage of having no maximum unambiguous range limit, and when the pulse location within the PRI is randomized, there is no upper limit on the maximum unambiguous Doppler frequency shift. I feel the experiment results are sufficient. It was a pleasure reviewing this work and I can recommend it for publication in Sensors after a major revision. I respectfully refer the authors to my comments below.

1.         The English needs to be revised throughout. The authors should pay attention to the spelling and grammar throughout this work. I would only respectfully recommend that the authors perform this revision or seek the help of someone who can aid the authors.

2.         (References) Please adjust the style of all the references to meet the Sensors Journal requirement.

3.         (Page 9) The original figure 13 is not clear. Please redraw this figure clearly. Add some word in this figure, and indicate the imaging modules. The reviewer cannot see the Figs. 9-12, please provides this.

4.         (Section 1 Introduction) The reviewer hopes the introduction section in this paper can introduce more studies in recent years. The reviewer suggests authors don't list a lot of related tasks directly. It is better to select some representative and related literature or models to introduce with certain logic. For example, the latter model is an improvement on one aspect of the former model.

5.         The reviewer suggests to add a new section “related work” to classify the references into several groups.

6.         Experimental pictures or tables should be described and the results should be analyzed in the picture description so that readers can clearly know the meaning without looking at the body.

7.         (Table 1) All the values in this table should be with same data accuracy. The number of data after the decimal point are the same. Please check other Tables. The table 1 needs three lines at least.

8.         The authors are suggested to add some experiments with the methods proposed in other literatures, then compare these results with yours, rather than just comparing the methods proposed by yourself on different models.

9.         Discuss the pros and cons of the proposed model.

Reviewer 2 Report

There are some comments in the following for reference:

1. The TITLE in the report form is totally different from the one in the manuscript. Please choose one and maintain the same for these two places, of which the latter one seems better;

2. Please give the full name of "PRI" (Pulse Repetition Interval) when it is first shown in the abstract. Also for "AES-192" in the Introduction Line 27;

3. How to explain that "In this work only a single input/output antenna is required." in Line 34 and why was that? There should some explanation about the current problem for the previous research and a brief introduction for the work in this manuscript in-between this conclusion;

4. Could you please explain more about why choose BPSK not QPSK, 8PSK as the potential sequences or doing the comparisons?

5. There are missing content for the key figure 2-4, and 6-12;

6. For the missing part mentioned in 5, sorry I could not continue to judge the manuscript and do the review.

Reviewer 3 Report

The authors investigate the use of encrypted pseudo-random binary phase shift (AES-192) sequences for radar signal modulation to mitigate Doppler and range ambiguities. This is interesting research. Unfortunately, it seems most of figures are missing in the upload pdf file. I suppose the authors need to resubmit the manuscript for review.

Round 2

Reviewer 1 Report

The revised manuscript is improved compared to the former version. My previous comments are well addressed, and the presentation is improved significantly. The composition pattern and some other ideas are well elaborated, making them clearer. Overall, I tend to accept this manuscript.

Reviewer 3 Report

Thank you for your revisions. The reviewer and AE are satisfied with them. However, there are still some issues with English, so please go over the manuscript carefully and engage assistance if needed for your final submission.